Seamless transformation from use case to sequence diagrams

Alyami Abdulrahman 1 2 am.yami@ju.edu.sa
Pileggi Salvatore Flavio 2
http://orcid.org/0000-0001-9287-5995 Sohaib Osama 2 3 Osama.Sohaib@uts.edu.au
Hawryszkiewycz Igor 2
1 Department of Information Systems, College of Computer and Information Sciences , Jouf University, Sakaka , Saudi Arabia
2 School of Computer Science, University of Technology Sydney , Sydney , Australia
3 School of Business, American University of Ras Al Khaimah , Ras Al Khaimah , United Arab Emirates
Al-Hadhrami Tawfik
Electronic publication date: 2023 Jun 22
Publication date: 2023
Volume: 9
Electronic Location ID: e1444
Received 2023 Mar 31; Accepted 2023 May 26
Copyright: © 2023 Alyami et al.
Copyright year: 2023
Copyright holder: Alyami et al.
License: This is an open access article distributed under the terms of the Creative Commons Attribution License, which permits unrestricted use, distribution, reproduction and adaptation in any medium and for any purpose provided that it is properly attributed. For attribution, the original author(s), title, publication source (PeerJ Computer Science) and either DOI or URL of the article must be cited.
License URL: https://creativecommons.org/licenses/by/4.0/

Keywords: UML, System design, Requirements engineering

Funding: The authors received no funding for this work.

==============================
System design is an essential subject taught in information systems and has become a core course in its curriculum. Unified modelling language (UML) has been broadly adopted, and it is common to support the system design process using different diagrams. Each diagram serves a purpose by focusing on a specific part of a particular system. Design consistency ensures a seamless process, as the diagrams are generally interrelated. However, creating a well-designed system takes a lot of work, especially for university students with work experience. To overcome this challenge, aligning the concepts across diagrams is essential, which can help achieve better consistency and management of the design system, especially in an educational setting. This article is an extension of our previous work, as we have discussed a simple scenario of Automated teller machines to demonstrate the alignment concepts between UML diagrams. From a more technical perspective, the current contribution provides a Java program that aligns concepts by converting text-based use cases to text-based sequence diagrams. Then, the text is transformed in PlantUML to generate its graphical representation. The developed alignment tool is expected to contribute to helping students and instructors during the system design phases to be more consistent and practical. Limitations and future work are presented.

Introduction

System design is essential in computer science and information systems (IS). It has become a core subject taught in the information systems curriculum for undergraduates and schools of business. System design is an approach established around the 1970s that addresses business needs and the technical issues related to software development (Siau et al., 2021). System design is typically considered a collection of procedures to define the various system elements and components that adhere to a particular set of requirements (Hoffer, George & Valacich, 2013). While system analysis includes a process of immersion and understanding of the users’ experience toward a specific system to improve or design a new approach based on the required specifications (Alan, Barbara & David, 2002). Thus, system analysis and design are two topics merged into one course in IS (Kohli & Gupta, 2002). Teaching in such a discipline could be a bit difficult (Chen, 2005; Cybulski & Linden, 2000), due to various factors, for instance, severe environmental changes, technological advancements, industry demands, and developing or shifting business trends. These factors definitely affect the students’ learning process, which necessitates enhancing and improving teaching methods. In fact, the scientific research and project outputs must be well-framed and plausible in genuine business environments. At a more educational level, the emphasis is on integrated skills and designing curricula to educate students to meet the demands of the present and the future workforce (Scott, 2009; Holdsworth et al., 2008; Mustaquim & Nyström, 2013; Penzenstadler et al., 2018). Education is seen as a lifelong process (Blossfeld & von Maurice, 2011), which needs to be continuously developed in line with evolving personal and professional needs, including but not limited to determining reality, transferring values, and socializing learners so that they can contribute to their social development and knowledge advancement. In essence, system design and analysis are complementary processes. System analysis works at understanding user experience and input, which can be used to inform the design process, in contrast to system design, which is focused on developing a blueprint for the system. In order to ensure that the system satisfies the requirements and is successful in addressing the demands of the users, a mix of these two approaches is essential. As a result, the system development process for any system must include both system design and analysis.

Generally, there are two standard modelling approaches in system analysis and design: (i) the traditional method, known as the structured and (ii) the object-oriented (Harris et al., 2006). The former consists of two phases, i.e., analysis of the system and then designing it by using a number of diagrams such as data flow diagrams and entity-relationship diagrams. The latter is commonly understood as data-centric by using unified modelling language (UML), which is a set of entities, namely “classes” that encapsulate the data known as “attributes” as well as the processes “methods” related to every entity (Siau et al., 2021). UML is a language that was proposed in the 1990s and adopted in practice for modelling software requirements (Bucchiarone et al., 2020). UML is also seen as a set of different approaches, such as object-oriented notations known as object-oriented design, object modeling technique, and object-oriented software engineering (Gomaa, 2006). UML has been broadly adopted in the teaching system analysis and design (Burton & Bruhn, 2004; Tanner & Scott, 2015), which combines different diagrams to represent a system’s behaviors and features. Typically, a single diagram is a graphical representation of a particular part of the target system. Ultimately, a system model includes several diagrams to illustrate the target design.

Furthermore, the model might incorporate or connect to additional descriptions or documentation related to different stages of the development process (Baumeister et al., 2003; Duursma, Olsson & Ulf, 1993). However, the link among these stages by multiple diagrams should be more consistent regarding the adopted concepts to provide better system management and a seamless process. More specifically, some UML diagrams share standard information in terms of concepts and elements. The outcome of a diagram can be an input for another diagram. Accordingly, considering the design consistency among the relevant UML diagrams results in better teaching system design and seamless outcomes (Harris et al., 2006), which supports system development and design whether in learning, teaching or systems design.

This article is an extension of our previous work (Alyami, Pileggi & Hawryszkiewycz, 2021), as we have discussed and analyzed a simple scenario of (ATM) to demonstrate the alignment of concepts between UML diagrams (i.e., use case and sequence diagrams) theoretically. Our research goal in this article is to use concept alignment to establish consistent system design across UML diagrams. This study has the potential to make a substantial contribution to the field of system design by establishing a consistent approach to UML diagrams and aligning concepts. Increased output, better teamwork, and a decrease in development-related errors are all advantages of using a consistent design approach. In general, this research goal has the potential to significantly advance the study of system design and its real-world applications. Transformation of the use case diagram into a sequence diagram has already been addressed in earlier and recent previous studies (Alami, Arman & Khamayseh, 2020; Khan & Mahmood, 2016; Murti, 2022; Sarma, Kundu & Mall, 2007; Sawprakhon & Limpiyakorn, 2014; Swain, Mohapatra & Mall, 2010; Thakur & Gupta, 2014). However, the difference is that this work takes the same perspective but considers the alignment of concepts between relevant diagrams. We propose a new way and easy seamless process of doing the transformation. We have used a text-based diagram generation language known as PlantUML (Brown, 2020), rather than using the traces in the extensible markup language (XML) format like previous approaches (Conrad, Scheffner & Freytag, 2000; Rambhia, 2002). This work focuses on text-to-text-based transformation using aligning concepts. In other approaches, transformation is based on traversing the diagram element by element, designed graphically and generated with XML format. A Java program is developed to convert text-based use cases to the text-based sequence diagram. Then, the text is transformed in PlantUML to generate its graphical representation. During the system design process, turning a use case diagram into a sequence diagram can be helpful. This is due to the fact that it permits a more in-depth examination of the relationships between the different system components, which then identifies potential for improving resource efficiency and minimizing environmental impact. In order to identify opportunities for improving resource consumption and reducing environmental impact, use case and sequence diagram alignment can offer a more thorough picture of how the system’s actors and components interact.

The article is structured into five sections. The second section reviews previous research on the subject, while the third section explores the alignment concepts between the target diagrams. The fourth section outlines the methodology employed in conducting the study, and the fifth section provides an overview of how the PlantUML tool works. Finally, the sixth section details the implementation process and presents the findings of the alignment tool, followed by the conclusion.

Related work

According to Jyothi & Rao (2011), UML is a fundamental modeling language that is both robust and versatile. It enables specialists to use diverse diagrams during various system development process phases, improving the process as a whole. The most well-known UML diagrams were assembled by Reggio et al. (2013) using data from a number of books, online tools, courses, and training programs. The target system is designed using these UML diagrams, which concentrate on distinct traits and facets. To adapt relevant information and spread it throughout several stages of the development process, it is frequently required to trace or align these diagrams (Nistala & Kumari, 2013). Several studies (Barmi, Ebrahimi & Feldt, 2011; Nistala & Kumari, 2013; Sousa & Do Prado Leite, 2014) suggest that traceability and alignment are largely interchangeable terms in the context of software development. The term “traceability” often refers to recording the flow of information (Jyothi & Rao, 2011; Kirova et al., 2008). Beyhl, Berg & Giese (2013) explored the benefits of traceability in innovative engineering processes, as it can aid in the successful execution of new concepts for products and services. Jyothi & Rao (2011) proposed various tracing strategies for extreme programming (Lindstrom & Jeffries, 2004) and scrum methodologies (Permana, 2015), some of which involve requirements. Agile methodologies place great importance on the model development process, while traceability and alignment are critical for developing more productive system models (Molenaar et al., 2020). Additionally, seamless alignment concepts are relevant for model transformation within the agile development process (Jyothi & Rao, 2011; Marlowe & Kirova, 2008).

There are various methods for model transformation suggested by Selonen, Koskimies & Sakkinen (2003). Memon et al. (2019) create a method for moving data from UML class diagrams to another model with a focus on idea traceability. Additionally, Hue et al. (2019) and Hue, Hanh & Binh (2018) offer a technique for automatically and methodically generating test cases from developed use cases. Yoshino & Matsuura (2020) offer a method for identifying the specifications that must be included in computer programs in order for them to automatically transfer data from an activity diagram and produce a sequence diagram. A method for automatic model transition has also been put forth in Ramesh, Kanth & Rao (2016). The process of data transformation used rules to develop an entity relationship diagram and create structured query language from unified modeling language class diagram. Segundo, Herrera & Herrera (2007) suggest a technique for teaching students and fresh analysts how to create sequence diagrams based on descriptions provided by a natural language. Moreover, Souza et al. (2015) apply an approach known as the semi-automatic transformation, which is supported also by a number of transformation criteria. Furthermore, Yue, Briand & Labiche (2010) focuses on developing traceability connections between the system requirements and the created diagrams. Traceability is important in software development because it helps engineers comprehend the connections between different software system artifacts (Molenaar et al., 2020). A technique suggested in Khan & Mahmood (2016) to facilitate and streamline the transition from need to artifact design. The design thinking methodology can also be used in the innovation process to foster creativity (Alyami & Hawryszkiewycz, 2020; Beyhl, Berg & Giese, 2013). Since concepts from diverse tools must be linked to give a uniform design in this circumstance, alignment and traceability are extremely crucial. Liu, Xu & Zou (2018) list two significant benefits of tracing the requirements: (i) it provides direction while making model changes, and (ii) it enhances how the final model is presented to users. The importance of idea alignment across UML diagrams in the context of education with a focus on system design hasn't been specifically discussed in the literature, to the authors’ knowledge.

Alignment of concept perspective

Concept alignment is the recognition and definition of semantic equivalences between concepts from various diagrams so that they can be interpreted as a consolidated knowledge source for the design. The design process improves the seamlessness of the target system by matching the adopted concepts directly. Syntactic alignment and semantic alignment are the two types of alignment that can be distinguished (Branigan, Pickering & Nass, 2003; Brockmans et al., 2006). Semantic alignment refers to inexplicit mapping by focusing on equivalent concepts among the different components; for example, actors in a use case diagram can be the exact as objects in a sequence diagram. However, the interactions among the objects in a sequence diagram can be aligned indirectly with the identified description of the use case diagram.

In our previous work (Alyami, Pileggi & Hawryszkiewycz, 2021), we emphasized two diagrams (i.e., use case and sequence). We discussed an example to emphasize the importance of aligning concepts during the system design process. Many studies from our previous literature published in Harris et al. (2006) illustrated that the consistency between use case and sequence diagrams facilitates a better system design. Additionally, other contributions endorsed the consistency of such diagrams, reported in Alami, Arman & Khamayseh (2020), Khan & Mahmood (2016), Sarma, Kundu & Mall (2007) and Sawprakhon & Limpiyakorn (2014). Accordingly, we have developed a syntactic alignment between use case and sequence diagrams to help students and instructors during the system design phases. We believe the syntactic alignment has contributed to speeding up the process of development, which eventually supports the design to be more consistent and effective.

At a general level, alignment is more about using the basic terms from the use case and mapping them in a sequence diagram. A use case diagram typically includes different essential elements: Use cases represent characteristics needed in the target system.

Each use case indicates further descriptions of the scenario.

An actor who triggers use cases.

A communication line is a connection between an actor and the use case.

In contrast, a sequence diagram contains objects and communication sequences. Sequences are collections of organized interactions among several objects. These interactions are numbers that describe how the system’s stages should proceed. Sequence diagrams are often the realization of use cases in sequential order for the system being created. As a result, Table 1 provides the direct alignment from the use case diagram to the sequence diagram in terms of syntax.

Table 1 Alignment of use case and sequence diagrams.

Use case diagram	Sequence diagram	
Actors	Objects	
Associations	Interactions	
Links	Directions of interactions	

Once more, in our previous work, we illustrated the process of the alignment concepts with a simple case study in detail as clients use an ATM system to carry out the transition from their bank accounts. The ATM scenario is a common example that has been adopted in previous studies (Alami, Arman & Khamayseh, 2020; El-Attar, 2011; Haugen et al., 2005; Panigrahi et al., 2018; Ullah, Faiz & Haleem, 2022; Whittle & Schumann, 2000). Nevertheless, Figs. 1 and 2 depict the final outcome of the aligned concepts from the use case to sequence diagrams. These figures are recalled from our previous work with a minor modification to meet the implementation requirements (Alyami, Pileggi & Hawryszkiewycz, 2021).

Figure 1 Customer’s use cases and interactions with the ATM system.

Figure 2 Created sequence diagram based on use cases.

Following the alignment principles defined in Table 1, Fig. 1 shows the use cases and the interaction between the actor “customer” and the ATM system. Looking at the first use case, “Validate the Customer” has the association steps (i.e., insert card, prompt pin code and enter pin code). The actors of the use case are considered the objects of sequence diagrams, while the association steps in the use case are set as the interactions of the sequence diagram. Then the links of the use cases are represented as the direction of the interaction in the sequence diagram, as shown in Fig. 2. Ultimately, the sequence diagram in Fig. 2 is generated from the use cases in Fig. 1 by following the exact alignment principles. More descriptions can be found in our previous work (Alyami, Pileggi & Hawryszkiewycz, 2021), as we have generated a single sequence diagram for each use case presented in Fig. 1.

The following section presents the methodology of the current contribution, which focuses more on implementing the alignment tool by the Java program and the PlantUML to generate the graphical representation.

Materials and Methods

This section presents the methodology and the process adopted to implement the alignment tool. The tool is developed using three open-source software. These are Java programming language (Oracle, 2012), Eclipse modeling framework (Eclipse Foundation, 2015) and PlantUML (PlantUML, 2015).

Java is a general-purpose and high-level object oriented programming (OOP) language (Oracle, 2012). Java lets the programmers write once and run anywhere (WORA), meaning it needs to be compiled once and executed on all platforms that support Java (Kramer, Joy & Spenhoff, 1996). The platforms need a Java virtual machine (JVM) (Craig, 2006). JAVA language does not come with its IDE for development. Eclipse modeling framework is used as an IDE for Java development (Geer, 2005).

Eclipse is an integrated development environment (IDE) that supports several programming languages for development. It comes with several plugins for customizing the environment. This IDE is written in Java and is widely used by Java developers. It supports several other languages (Kastner et al., 2009), including but not limited to Ada, ABAP, C, C++, PHP, Perl, Prolog, Python, and Mathematica. Here, Eclipse IDE is used for writing Java code. Eclipse is chosen here because it supports both Java and PlantUML software tools and is lightweight.

PlantUML is used to code the use case diagram textually by following the syntax of the language (Brown, 2020). PlantUML is also used to generate the graphical representation of the diagrams. Java is a general-purpose object-oriented programing language. It codes the transformation from the use case to the sequence diagram. PlantUML is used to create diagrams from plain text. PlantUML is also used to allow blind people to design UML diagrams (Muller, 2012). It is also termed a domain-specific language. PlantUML is developed in Java language and comes with the Eclipse Plugin.

At a methodological level, this study achieves the seamless transformation based on traversing the diagram element by element, designed graphically and generated with XML format. A Java program is developed to convert text-based use cases to the text-based sequence diagram. Then the text is transformed in PlantUML to generate its graphical representation. The approach involves a step-by-step traversal of the diagram elements, initially designed graphically and generated in an XML format. The transformation process involves a Java program that converts text-based use cases to text-based sequence diagrams, followed by the transformation of the text using PlantUML to generate a graphical representation. A methodical approach to diagram transformation ensures that the process is repeatable, consistent and accurate. Traversing the diagram element by element ensures that no information is lost during the transformation process. This is particularly important when dealing with complex diagrams with many interconnected elements. Developing a Java program to convert text-based use cases to text-based sequence diagrams provides a standardized approach to the transformation process. This approach reduces the manual effort required to create the diagrams and ensures that the transformation process is consistent and reliable. It also enables the automation of the transformation process, which can further increase efficiency.

Using PlantUML to generate the graphical representation of the diagrams ensures that the resulting diagrams are consistent with industry standards. PlantUML provides a range of diagramming features that can be customized to suit the project’s specific requirements. Additionally, the resulting diagrams can be easily shared and reviewed by stakeholders, ensuring that everyone involved in the project clearly understands the system design.

The following section explains the process of generating the graphical representation of a use case diagram and a sequence diagram using PlantUML.

Plant uml

While preparing documents, using a word processor like Microsoft Word or opt for LaTeX is always a choice. Microsoft works like what you see is what you get. However, LaTeX functions embedding text within the list of commands and compiling it will yield the document. In Microsoft Word, a small change can affect the document’s formatting, while in LaTeX, the focus can only be kept on the content rather than formatting. A similar choice we get in using the PlantUML tool. UML diagrams are often developed using graphical tools, including but not limited to Microsoft Visio, Rational Rose, StarUML, and Enterprise Architects (Khaled, 2009). All such tools use the drag and drop of elements for building diagrams. PlantUMl is a tool used to create UML diagrams from plain text language. In PlantUML, UML diagrams are written down instead of drawn from elements. The language used to write these diagrams can be called application-specific because it only works with PlantUML. It can also be described as a scripting language for UML diagrams. PlantUML is an open-source tool, and it also comes with plug-ins to be added to several popular platforms and tools such as Eclipse, Net Beans, LaTeX, Microsoft Word, and PHP.

In the following section, we explore some basic syntax of the PlantUML coding language to understand how it works. First, each block of the code has to start and end statements @startuml and @enduml, respectively. These two statements are common for all diagrams that are being designed. The code between these two statements contains the code of the relevant diagram for which we shall write the statements (Madanayake, Dias & Kodikara, 2017).

The following subsections outline the basic syntax of creating the use case diagram, followed by the sequence diagram.

Use case diagram

Elements of the use case diagram can be expressed as follow:

Use cases

Use cases can be defined with the keyword ‘usecase’ or by just enclosing within the parenthesis without mentioning the keyword ‘usecase’. Furthermore, usecase can be defined with an alias by writing the keyword with ‘as’ followed by the alias you want it to be associated with. This alias is used while describing the relationship with the actors. A sample code is given in Fig. 3A, while the diagram generated from the code is shown in Fig. 3B.

Figure 3 Use case names scenario.

(A) Sample code. (B) Diagram generated from the code.

Actors

An actor can be defined by either enclosing within colons or can be prefixed with the keyword ‘actor’. Similar to use cases, it can also be specified with an alias by appending the word ‘as’ followed by the alias you want it to be associated with. This alias is used while defining the relationship with the use cases. A sample code is given in Fig. 4A, while the diagram generated from the code is displayed in Fig. 4B.

Figure 4 Use case actors.

(A) Sample code. (B) Diagram generated from the code.

Different styles can represent actors in the diagram. The default style is ‘Stick man’; other types include ‘Awesome man’ and ‘Hollow man’.

Use case description

Use cases can be added with a description that spans across several lines. Quotes can be used to add the description. Separators can separate description. Following are the type of separators supported by the PlantUML: ‐‐(dashes)

.. (periods)

== (equals)

__ (underscores)

Figure 5B, shows the generated diagram from the code presented in Fig. 5A.

Figure 5 Use case description.

(A) Sample code. (B) Diagram generated from the code.

Relationships

The symbol of an arrow ‐‐> is used to define the relationship between actors and use cases. The more dashes in the arrow, the longer will be the size of the arrow. The link between the actor and the use case can be set as vertically or horizontally in the diagram. A sample code is given in Fig. 6A, while the generated diagram from the code is shown in Fig. 6B.

Figure 6 Relationships.

(A) Sample code. (B) Diagram generated from the code.

The direction and type of the link between the actor and the use case can be changed from left to right. Furthermore, the style of the arrow can also be changed from a continuous line to a dotted line. A sample code is given in Fig. 7A, while the generated diagram from the code is shown in Fig. 7B.

Figure 7 Direction and type of the relationship.

(A) Sample code. (B) Diagram generated from the code.

Apart from these elements, there are other elements supported by PlantUML, for instance, package, extension, notes, stereotypes, and splitting diagrams. However, these are not objects of this work. Further information can be referred to PlantUML guide (Sasidharan, 2016).

The following section is dedicated to illustrating the elements of the sequence diagram.

Sequence diagram

In the main, the sequence diagram consists of participants and messages. Participants are not declared explicitly in the sequence diagram as actors, which are displayed in the use case diagram. For messages, directed arrows -> and double dashed arrows ‐‐> are used. A Sample code is given in Fig. 8A, and the generated diagram from the code is presented in Fig. 8B.

Figure 8 Sequence diagram.

(A) Sample code. (B) Diagram generated from the code.

Declaring participant

Declaring the participant is optional. However, declaring it will provide more power and control to it. For example, the participant’s shape can be changed from default to its corresponding user class. These shapes include actor, boundary, control, entity, database, collections and queue. Participants can be renamed using ‘as’ keyword.

Message to self

A message can be sent from one participant to another and to itself. At the same time, it can include multi-line messages.

Message sequence number

Every message that is sent from one participant to another must be numbered. There are two ways to number the message. Either it can be given manually, or it should be autogenerated to add numbers. The keyword used is called ‘autonumber’. A sample program is given in Fig. 9A, and the generated diagram from the code is shown in Fig. 9B.

Figure 9 Message sequence number.

(A) Sample code. (B) Diagram generated from the code.

Implementation and results

The complete internal structure of the tool containing the alignment process of use case to sequence diagram is illustrated in Fig. 10. The alignment tool mainly consists of four steps. The first step is to program the use case diagram in plain text. The second step is generating a graphical representation of the use case diagram from plain text. This second step is optional, not mandatory and can be omitted. However, it is necessary to visualize the diagram graphically to verify whether it is correctly programmed. The third step is generating a text-based sequence diagram from a text-based use case diagram. This step is the core of the alignment process. Finally, the fourth step is developing a graphical representation of the sequence diagram from the text generated in the third step.

Figure 10 The internal structure of the tool.

A further explanation of each step is explained one by one in the following sections.

Step 1: write the text for the use case diagram

The first step of the alignment tool is to program the use case diagram in PlantUML text-based language. The same case study of the ATM scenario is used to test our theoretical alignment concept addressed in previous work and summarized in “Related Work”.

Here the use case diagram is coded in plain text for the same running example of the ATM scenario. The complete program is given below. @startuml

left to right direction

actor Customer

actor ATM

actor BankSystem

usecase UC1 as “1.VALIDATE THE CUSTOMER.

==

<size:10>Insert Card</size>

‐‐

<size:10>Prompt Pin Code </size>

‐‐

<size:10>Enter Pin Code </size>”

Customer‐‐>UC1

UC1<‐‐ATM

usecase UC2 as “2.AUTHENTICATION.

==

<size:10>Check Pin Code </size>

‐‐

<size:10> Reply </size>

‐‐

"

ATM‐‐>UC2

UC2<‐‐BankSystem

usecase UC3 as “3.CHECK BALANCE.

==

<size:10>Prompt Access Status </size>

‐‐

<size:10>Check Amount </size>

‐‐"

Customer‐‐>UC3

UC3<‐‐ATM

usecase UC4 as “4.CHECK AMOUNT.

==

<size:10> Check Database </size>

‐‐

<size:10> Reply </size>

‐‐”

ATM‐‐>UC4

UC4<‐‐BankSystem

usecase UC5 as “5.TRANSFER FUNDS.

==

<size:10> Prompt Amount </size>

‐‐

<size:10> Transfer Funds </size>

‐‐”

Customer‐‐>UC5

UC5<‐‐ATM

usecase UC6 as “6.PERFORM TRANSFER.

==

<size:10> Update Database </size>

‐‐

<size:10> Transfer Confirmation </size>

‐‐”

ATM‐‐>UC6

UC6<‐‐BankSystem

@enduml

Line 1 is the beginning of the program. Line 2 enforces the left-to-right direction of the arrow instead of the default behavior of top to bottom. Lines 3 to 5 are the declaration of the actors. From lines 6 to 14 are the first use case of “VALIDATE THE CUSTOMER”. Line 6 is the declaration of the use case. Lines 7 to 12 are the links of use cases. Lines 13 and 14 are the use case links with the actors: Customer and ATM. Lines 15 to 23 are the second use case “AUTHENTICATION”. Lines 24 to 31 are the third use case “CHECK BALANCE”. Lines 32 to 39 are the fourth use case “CHECK AMOUNT”. Lines 40 to 47 are the fifth use case “TRANSFER FUNDS”. Lines 48 to 55 are the sixth use case “PERFORM TRANSFER”. Finally, line 56 is the end of the program.

Step 2: create a use case diagram from the code

The code of the use case diagram is explained in the first step, which would be written in a simple notepad or any WordPad file and then saved as a text file with a .txt extension (e.g., usecase.txt). To create the graphical representation of the text file, a PlantUML Java batch file is executed. PlantUML batch file can be downloaded from PlantUML’s official website or from several websites that allow creating the diagrams online, such as https://www.plantuml.com/plantuml/uml. The PlantUML batch file and the use case diagram text should be in the same folder. The next step is to right-click the batch file and click edit. A new editable window would appear. Just write the following command in the file ‘java-jar plantuml.jar usecase.txt’, followed by saving and closing the file, as shown in Fig. 11.

Figure 11 The process of using PlantUML.

Afterward, double-click the batch file, and within a few seconds, a new PNG file will appear in the same folder containing the PlantUML batch file and usecase.txt file. That PNG file would be the graphical representation of the use case diagram. For the code given in the first step, the following PNG file is created, as shown in Fig. 12. As mentioned earlier, Eclipse IDE also comes with the PlantUML plugin. Therefore, the graphical representation can also be viewed in the IDE. However, the authors did not find the plugin to be stable while implementing this tool, and encountered problems.

Figure 12 Graphical use case diagram generated by PlantUML for the ATM scenario code.

Step 3: alignment from use case text to sequence text

The third and most important step is the generation of sequence diagram code from the use case diagram code given in step 1, considering the perspective of aligning concepts between the target diagrams. A visual illustration of the tool is presented in Fig. 13.

Figure 13 Generation of sequence diagram code from use case diagram code.

The alignment tool is implemented in Java, which takes the usecase.txt file as input and generates the corresponding sequence diagram text file ‘sequence.txt’ output. The alignment code is programmed in Eclipse IDE that follows alignment principles as explained in previous work and recalled in “Related Work”. After executing the Java file, it automatically generates the following code below. @startuml

Customer->ATM: 1. Insert Card

ATM->Customer: 2. Prompt Pin Code

Customer->ATM: 3. Enter Pin Code

ATM->BankSystem : 4. Check Pin Code

BankSystem ->ATM: 5. Reply

Customer->ATM: 6. Prompt Access Status

ATM->Customer: 7. Check Amount

ATM->BankSystem: 8. Check Database

BankSystem->ATM: 9. Reply

Customer->ATM: 10. Prompt Amount

ATM->Customer: 11. Transfer Funds

ATM->BankSystem: 12. Update Database

BankSystem->ATM: 13. Transfer Confirmation

@enduml

In this code, line 1 is the beginning of the program, and from Lines 2 to 14 are all interactions of the use cases that have now become the messages between the objects. Line 15 is the end of the program. Contrary to the use case diagram, messages are encapsulated in the body of each use case, followed by the actors who interact with that use case. In the sequence diagram, each interaction is specified with the objects that belong to it, as well as its direction from where it initiates and ends.

The pseudocode of this algorithm is listed in Table 2. It starts with the initialization of variables corresponding to the use case file, sequence file and an array of strings. The array of strings is used to store exact relevant elements of the use case in order to use them for sequence diagram generation. Afterwards, a loop is applied to read the usecase.txt file line by line until the file ends. The algorithm uses the traceability mechanism for each use case until it finishes all the use cases before the file ends. The algorithm searches for the string use case in the file and records the start of the use case. Then it searches for the <size:10> and </size> strings to record the associations (links) of that use case in the array of strings after storing all the associations of use cases. It then looks for ‐‐> string and <‐‐ string, respectively, to extract the exact actors performing that use case and their left, and right position of the string is correctly recorded. This process is repeated until the end of the file is reached, and all the information is recorded in the string.

Table 2 Algorithm to transform use case diagram to sequence diagram.

 Input: Usecase.txt file that contains the code in PlantUML of Use case diagram	
 Output: Sequence.txt file that contains the code in PlantUML sequence diagram	
1  Initialization of variables: readfile = usecase.txt, writefile = sequence.txt and string array interaction	
2  write @staruml in the beginning of interaction array	
3  while (readfile.readline() not equal to null) do	
4   read one line from the usecase.txt file	
5   search the string usecase in the line	
6   try	
7    extract usecase from the line	
8   catch	
9   search the string for <size:10> and </size>	
10   try	
11    extract Interaction between <size:10> and </size>	
12    store in interaction array	
13   catch	
14   search the string for ‐‐>	
15   try	
16    if string contains ‐‐> then	
17     extract left side string of ‐‐>	
18     store in interaction array	
19    else do nothing	
20   catch	
21   search the line string for <‐‐	
22   try	
23    if string contains <‐‐ then	
24     extract right side string of <‐‐	
25     store in interaction array	
26    else do nothing	
27   catch	
28  end	
29  create the sequence.txt file in writefile	
30  write @staruml in writefile	
31  for (each use case in the interaction string)	
32   write left side string of ‐‐> in the writefile	
33   write the symbol ‐‐> in the writefile	
34   write right side of string of ‐‐> in the writefile	
35  end for loop	
36  write @enduml in the end of writefile	
37  close writefile	

The next step is to generate a sequence diagram by extracting the information from the array of strings and putting it into the sequence.txt file accordingly. Another loop is used until the end of the array of strings. For each use case, the association becomes the interaction and left-side and right-side strings stored in the array become the objects of the sequence diagram. This process is repeated until the end of the string. Finally, the @endUML string is added to the sequence diagram to close the diagram, which is the end of the algorithm.

Step 4: creating sequence diagram from the generated code

The last and final step of the tool is to visualize what was generated by our java program in the third step. When the tool yields the sequence.txt file, similar to the first step, the PlantUML batch file will create the diagram. The exact process will be applied as previously used to generate a use case diagram graphical representation. The PlantUML batch and sequence diagram text files should be in the same folder. The next step is to right-click the batch file and click edit. A new editable window will appear. Write the following command in the file ‘java-jar plantuml.jar sequence.txt’, then save and close it. Afterward, double-click the batch file, and within a few seconds, a new PNG file will appear in the same folder containing the PlantUML batch file and sequence.txt file. That PNG file will be the graphical representation of the sequence diagram. The following PNG file is created for the code generated in the third step, as shown in Fig. 14.

Figure 14 Sequence diagram created by PlantUML for the code generated by the alignment tool.

Conclusion

The article highlights the use of PlantUML, a text-based diagram generation language, for the text-to-text-based transformation of use case diagrams to sequence diagrams. This approach differs from previous approaches that relied on XML format traces and focused on the element-by-element traversal of graphical diagrams. Instead, this work aligns concepts between text-based use cases and text-based sequence diagrams. The contribution of this study includes, firstly, it allows for a more flexible and customizable approach to diagram creation as the constraints of graphical design tools do not limit it. Secondly, it facilitates better collaboration and easier management of changes.

Moreover, text-based transformation enables the creation of diagrams in multiple formats, making them more accessible to a broader range of users. Using a Java program to convert text-based use cases to text-based sequence diagrams provides a standardized approach to diagram creation, which can be easily automated, reducing manual effort and increasing efficiency. Aligning the concepts of use case and sequence diagrams in a text-based format allows for more accessible translation between the two formats, as they share common elements such as actors, actions, and messages. This approach also allows for better integration with other text-based software engineering tools, such as code generation tools.

In conclusion, the text-to-text-based transformation using aligning concepts of use case diagrams to sequence diagrams using PlantUML is a promising approach that offers several benefits over previous methods. Its flexibility, standardization, and integration with other software engineering tools make it valuable for diagram creation and transformation.

In the educational context, aligning concepts between the relevant diagrams helps strengthen the system specification requirements, resulting in a seamless process. Additionally, the alignment concepts offer a more unified approach that enables better control of the system complexity over the different phases of the target design. Overall, the aligning concepts among homogeneous diagrams should be considered for providing a seamless process, which is the focus of our current study.

The limitation of this work is that certain features of the diagrams of both use cases and sequences need to be implemented. Our current focus is on the primary and essential features. Additionally, writing the code of the use case diagram needs to be followed in the same way explained in this article to avoid unnecessary errors. Future work will involve extending the alignment of the sequence diagram to the collaboration diagram. This is because the two diagrams have similar ideas and components, such as elements and objects (Dobing & Parsons, 2006). The collaboration diagram portrays objects and their relationships, illustrating how they interact with one another.

Supplemental Information

Supplemental Information 1 Use case code (java).

Click here for additional data file.

Supplemental Information 2 Use case code (text).

Click here for additional data file.

Additional Information and Declarations

Competing Interests

Author Contributions

Data Availability

Osama Sohaib is an Academic Editor for PeerJ Computer Science.

Abdulrahman Alyami conceived and designed the experiments, performed the experiments, analyzed the data, performed the computation work, prepared figures and/or tables, authored or reviewed drafts of the article, and approved the final draft.

Salvatore Flavio Pileggi conceived and designed the experiments, analyzed the data, authored or reviewed drafts of the article, and approved the final draft.

Osama Sohaib performed the experiments, analyzed the data, prepared figures and/or tables, authored or reviewed drafts of the article, and approved the final draft.

Igor Hawryszkiewycz conceived and designed the experiments, authored or reviewed drafts of the article, and approved the final draft.

The following information was supplied regarding data availability:

The text code for the scenario discussed in the article is available in Supplemental File.

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
