# Peer review of "Seamless transformation from use case to sequence diagrams"

_PeerJ Computer Science, doi:10.7717/peerj-cs.1444_

## Round 0.1 · original submission · Minor Revisions

The comments provided by the reviewers are required for this paper to be published in PeerJ Computer Science.

Reviewer 1 ·

Basic reporting

This paper discusses the alignment of concepts between UML diagrams, i.e., use case and sequence diagrams, which is relevant and important.

The introduction provides a comprehensive system analysis and design overview, including its history, importance, and modeling approaches. The paper highlights the significance of UML diagrams in teaching and designing systems, especially in the context of sustainability. The authors should discuss how sustainability relates to transforming use to sequence diagrams. Or remove the sustainability approach.

Experimental design

The section provides a clear and detailed description of the methodology and process adopted to implement the alignment tool. The author has effectively used various references to support their statements and provide a comprehensive understanding of the tools used in the implementation.
The use of Java programming language, Eclipse Modeling Framework, and PlantUML is well-justified and explained, and their roles in the implementation process are clearly delineated. The use of Eclipse as an IDE for Java development and PlantUML to generate graphical representations of the diagrams is an appropriate choice for this implementation.
Regarding the section that outlines the basic syntax of PlantUML, the author provides clear instructions and breaks down the process into manageable steps. Using references to support the content adds credibility to the text and makes it more convincing. Overall, the text is well-organized, easy to follow, and effectively conveys the key points about PlantUML. However, the author may consider adding more examples to illustrate how the syntax works in practice.

Validity of the findings

Overall, the paper presents an interesting approach to transforming use case diagrams to sequence diagrams using PlantUML, a text-based diagram generation language. The authors have highlighted the benefits of using this approach, such as flexibility, standardization, and integration with other software engineering tools. They have also discussed the potential educational applications of aligning concepts between homogeneous diagrams, which can help strengthen the system specification requirements and provide a more unified approach to system design.
However, there are a few areas that could be improved:

• While the authors have mentioned the benefits of their approach, it would be helpful to provide some examples or case studies to demonstrate the effectiveness of the proposed method.
• It is also unclear what the authors mean by "collaborative diagram." It would be helpful to provide more information on this and how it relates to the use case and sequence diagrams.

Additional comments

Overall, this paper provides a valuable contribution to the field of software engineering, and the proposed method has the potential to be useful in both educational and practical settings. However, addressing the above-mentioned points would strengthen the paper further.

Cite this review as

Reviewer 2 ·

Basic reporting

Comment 1: The abstract does not include any quantitative measurement; authors should consider updating the abstract accordingly.
Comment 2: Remove ‘algorithms’ from keywords.
Comment 3: One area where the Introduction could be improved is in the organisation and flow of the text. The paper jumps from discussing the importance of system analysis and design to discussing the traditional and object-oriented modeling approaches without a clear transition. Restructuring the text to have a clearer flow and organization would make it easier for readers to follow and understand the arguments being presented. In addition, include the research question and at least the objective of the study statements in the introduction.
Comment 4: Add one paragraph/bullet point at the end of the introduction; what you intend to do in this research and link them to respective section(s) of the manuscript. (contribution/s)
Comment 5: In Figure 14, the alignment algorithm's flowchart is unclear and not necessarily present.
Comment 6: Some figures without names/titles, please check all the figures.
Comment 7: Add Literature review section.
Comment 8: The conclusion needs to be rewritten, make it shorter and more precise.

Experimental design

Comment 1: The abstract does not include any quantitative measurement; authors should consider updating the abstract accordingly.
Comment 2: Remove ‘algorithms’ from keywords.
Comment 3: One area where the Introduction could be improved is in the organisation and flow of the text. The paper jumps from discussing the importance of system analysis and design to discussing the traditional and object-oriented modeling approaches without a clear transition. Restructuring the text to have a clearer flow and organization would make it easier for readers to follow and understand the arguments being presented. In addition, include the research question and at least the objective of the study statements in the introduction.
Comment 4: Add one paragraph/bullet point at the end of the introduction; what you intend to do in this research and link them to respective section(s) of the manuscript. (contribution/s)
Comment 5: In Figure 14, the alignment algorithm's flowchart is unclear and not necessarily present.
Comment 6: Some figures without names/titles, please check all the figures.
Comment 7: Add Literature review section.
Comment 8: The conclusion needs to be rewritten, make it shorter and more precise.

Validity of the findings

Comment 1: The abstract does not include any quantitative measurement; authors should consider updating the abstract accordingly.
Comment 2: Remove ‘algorithms’ from keywords.
Comment 3: One area where the Introduction could be improved is in the organisation and flow of the text. The paper jumps from discussing the importance of system analysis and design to discussing the traditional and object-oriented modeling approaches without a clear transition. Restructuring the text to have a clearer flow and organization would make it easier for readers to follow and understand the arguments being presented. In addition, include the research question and at least the objective of the study statements in the introduction.
Comment 4: Add one paragraph/bullet point at the end of the introduction; what you intend to do in this research and link them to respective section(s) of the manuscript. (contribution/s)
Comment 5: In Figure 14, the alignment algorithm's flowchart is unclear and not necessarily present.
Comment 6: Some figures without names/titles, please check all the figures.
Comment 7: Add Literature review section.
Comment 8: The conclusion needs to be rewritten, make it shorter and more precise.

Annotated reviews are not available for download in order to protect the identity of reviewers who chose to remain anonymous.
Cite this review as

---

## Round 0.2 · accepted · Accept

The paper is in an improved format, and is clearly suitable for publication in PeerJ Computer Science.

Reviewer 1 ·

Basic reporting

The author has addressed the previous comments effectively.

Experimental design

The author has addressed the previous comments effectively.

Validity of the findings

The author has addressed the previous comments effectively.

Cite this review as

Reviewer 2 ·

Basic reporting

Happy with the revised version

Experimental design

Happy with the revised version

Validity of the findings

Happy with the revised version

Cite this review as